# Behavioral Study of 3- and 5-Halocytisine Derivatives in Zebrafish Using the Novel Tank Diving Test (NTT)

**DOI:** 10.3390/ijms241310635

**Published:** 2023-06-25

**Authors:** Amaury Farías-Cea, Cristóbal Leal, Martín Hödar-Salazar, Erica Esparza, Luis Martínez-Duran, Irma Fuentes, Patricio Iturriaga-Vásquez

**Affiliations:** 1Laboratorio de Farmacología Molecular y Química Medicinal, Departamento de Ciencias Química y Recursos Naturales, Facultad de Ingeniería y Ciencias, Universidad de La Frontera, Temuco 4811230, Chile; a.farias01@ufromail.cl (A.F.-C.); c.leal13@ufromail.cl (C.L.); martinhodar@gmail.com (M.H.-S.); e.esparza01@ufromail.cl (E.E.); miguetzd@gmail.com (L.M.-D.); 2Programa de Doctorado en Ciencias de Recursos Naturales, Universidad de la Frontera, Temuco 4811230, Chile; 3Programa de Doctorado en Ciencias mención Biología Celular y Molecular Aplicada, Universidad de La Frontera, Temuco 4811230, Chile; 4Instituto de Ciencias Aplicadas, Facultad de Ingeniería, Universidad Autónoma de Chile, Temuco 4810101, Chile; irmafue@gmail.com

**Keywords:** novel tank diving test (NTT), anxiety, halogenated cytisine, zebrafish, nicotinic acetylcholine receptors (nAChRs)

## Abstract

Anxiety is a serious mental disorder, and recent statistics have determined that 35.12% of the global population had an anxiety disorder during the COVID-19 pandemic. A mechanism associated with anxiolytic effects is related to nicotinic acetylcholine receptor (nAChR) agonists, principally acting on the α4β2 nAChR subtype. nAChRs are present in different animal models, including murine and teleosteos ones. Zebrafish has become an ideal animal model due to its high human genetic similarities (70%), giving it high versatility in different areas of study, among them in behavioral studies related to anxiety. The novel tank diving test (NTT) is one of the many paradigms used for studies on new drugs related to their anxiolytic effect. In this work, an adult zebrafish was used to determine the behavioral effects of 3- and 5-halocytisine derivatives, using the NTT at different doses. Our results show that substitution at position 3 by chlorine or bromine decreases the time spent by the fish at the bottom compared to the control. However, the 3-chloro derivative at higher doses increases the bottom dwelling time. In contrast, substitution at the 5 position increases bottom dwelling at all concentrations showing no anxiolytic effects in this model. Unexpected results were observed with the 5-chlorocytisine derivative, which at a concentration of 10 mg/L produced a significant decrease in bottom dwelling and showed high times of freezing. In conclusion, the 3-chloro and 3-bromo derivatives show an anxiolytic effect, the 3-chlorocytisine derivative being more potent than the 3-bromo derivative, with the lowest time at the bottom of the tank at 1mg/L. On the other hand, chlorine, and bromine at position 5 produce an opposite effect.

## 1. Introduction

Anxiety is a serious mental disorder, categorized by the American Psychiatric Association in the Diagnostic and Statistical Manual of Mental Disorders (DSM-V) [1]. The World Health Organization (WHO) in 2015 reported a total of 264 million cases worldwide [2]. Santomauro et al. [3] estimated that the prevalence in 2020 of anxiety disorder cases was 298 million people, but this had to be recalculated due to the COVID-19 pandemic, with a new estimated prevalence of 374 million people. In recent studies, the global population, according to Xiong et al. [4] revealed relatively high rates of anxiety symptoms, 6.33–50.9%, while Delpino et al. [5] determined that 35.12% of people experiencing these, had an anxiety disorder during the pandemic.

Anxiety is associated with different mechanisms, and one of them is that of nicotinic acetylcholine receptors (nAChRs) which are pentameric ion channels, and they can be heteromeric or homomeric subtypes depending on their subunit composition (α2–α10 and β2–β4) and stoichiometry [6]. The nAChRs family plays an important role in regulating emotional behaviors [7], and some evidence has shown that nAChRs are involved in anxiety, depression and nicotine addiction [6]. The α4β2 subtypes, which are the major nAChRs present in the central nervous system (CNS) [6,7] have been proposed as the principal subtypes involved in anxiety and nicotine addiction; however, another nAChR subtype could contribute to these disorders. The nAChR subtypes have been studied in different animal models such as non-human primates, cats, frogs, rats, mice, and zebrafish, associated with different behaviors such as memory loss and drug abuse, among others, and play a key role in the anxiolytic activity of nicotine [8,9,10,11,12,13,14].

Zebrafish (*Danio rerio*) has been used as a model organism for the last 40 years, because these fish share approximately 70% of genes with humans, in addition to having the properties of fast development, high fecundity which provides a large number of individuals, cheap maintenance, and ease of manipulation, making it an ideal model in areas such as biological development, neurophysiology, biomedicine, and behavioral studies [15,16,17,18,19]. Its versatility in development and the number of individuals for experimentation make the zebrafish an appropriate animal model for studying new drugs [20,21]. Behavioral studies have been used with zebrafish in complex paradigms related to memory, anxiety, stress, addiction, and the reinforcing properties of drug abuse [11,22,23,24].

The novel tank diving test (NTT) is a model that is widely used by different research groups for studying swimming behavior associated with anxious or stressful states [25,26]. The novel tank diving test is conceptually similar to the rodent open-field test, as it takes advantage of the instinctive behavior of zebrafish and rats to seek refuge when exposed to unfamiliar environments. A fish feels anxious when it spends a longer time at the bottom of the tank, and this behavior allows us to determine if a new drug has an anxiolytic effect when the bottom dwelling time is decreased (Figure 1); however, there are any others parameters to observe such us latency, average velocity and freezing that as a whole can generate an anxiolytic profile of the behavior of zebrafish [27,28,29].

In the NTT paradigm, the swimming behavior of zebrafish has been studied using different drugs, such as nicotine, fluoxetine, caffeine, and cytisine among others [11,12,20,30,31,32,33]. These drugs have also been shown to interact with nAChRs [11,14,34,35,36]. Cytisine is an alkaloid present in plants of the genus Sophora (*Fabaceae*) and has shown partial agonist activity against nicotinic receptors, as well as showing anxiolytic effects on rats and zebrafish [33,37,38,39]. This alkaloid has been halogenated using different methods; however, the reactivity of the pyridone functional group only produces a substitution in the 3- and 5- position and at a lower proportion the 3,5-dihalogenated derivative is yield. These compounds have shown a high affinity (sub nM) for the nAChR subtypes; the 3-bromo derivative displays selectivity for the heteromeric α4β2 (IC50 = 0.30 nM) and α4β4 (IC50 = 0.28 nM) nAChRs over the homomeric α7 subtype (IC50 = 31 nM). Other positions, such as the 5-halo and the 3,5-dihalo derivatives, display low affinities in the microM range [40,41]. Bermudez et al. studied the functional activity of these halo derivatives in different nAChR subtypes overexpressed on *xenopus oocytes* using two-electrode voltage clamp experiments. Despite their high-affinity bindings, all of them display partial agonism with different efficacies and potencies based on the subtype composition, where position 3 appears to be the most potent pattern substitution [14,36,42].

Based on the data that show different affinities and potencies of the halo-cytisine derivatives, and point to 3-halo derivatives as the most potent ones compared to those of position 5, the main aim of this work was to determine and compare the anxiolytic effect of 3- and 5-halo-cytisine derivatives (Figure 1) using the novel tank diving test (NTT) as an anxiolytic paradigm and to confirm that nicotinic partial agonists exert anxiolytic effects in this model. 

## 2. Results

### 2.1. Drugs

Cytisine was isolated from seeds of *Sophora secundiflora*. Their structure was confirmed via one-dimensional 1H-NMR analysis. The hemisynthesis of 3-halocystisine and 5-halocystisine was carried out via halogenation with N-chlorosuccinimide (NClS) or N-bromosuccinimide (NBS) and acetic acid at 60% as a solvent [40]; for both reactions (chlorination and bromination) the yields were between 25 and 30% for each compound prepared. The chemical structure of the halocytisine derivates was stablished via one-dimensional 1H-NMR analysis and compared to that shown in the literature data [40]. NClS and NBS were purchased from AK Scientific (Union City, CA, USA) and glacial acetic acid was purchased from Winkler ltda (Santiago, Chile).

### 2.2. Effects of 3-, and 5-Halocytisine Derivatives on the Novel Tank Diving Test (NTT)

Adult fish were exposed to 3-chlorocytisine, 3-bromocytisine, 5-chlorocytisine or 5-bromocytisine (n = 10 per compound/dose) for 3 min and then maintained for another 5 min in a holding tank before starting the experiment on swimming behavior during the NTT for a period of 5 min. Halocytisine derivatives were used at four different concentrations: 1, 10, 25, and 50 mg/L.

#### 2.2.1. Bottom Dwelling Time

The bottom dwelling time (Figure 2) for the control group was 122.5 ± 20.51 s, and for the 3-chlorocytisine derivative this time was significantly reduced to 57.56 ± 10.25 s at 1 mg/L (~53%) and markedly reduced to 10.34 ± 3.02 s at 10 mg/L (~91%); however, the bottom dwelling time at higher concentrations (25 mg/L and 50 mg/L) resulted in an increase in the bottom dwelling time of 239 ± 30.46 s and 203 ± 36.94 s, respectively, which is about two times more than that of the control. For 5-chlorocytisine, the time spent at the bottom was increased by around -fold, at ~200 s on average, compared to that of the control at all concentrations used, except for 10 mg/L where the bottom dwelling time was significantly reduced to 49.16 ± 12.36 s (~59%). For the 3-bromocytisine derivative, the time spent at the bottom was not significantly different from that of the control at 1 mg/L, but a significant reduction in the bottom dwelling time was observed at 10 mg/L, 25 mg/L, and 50 mg/L with a time reduction of 15.82 ± 5.98 s (~87%), 16.58 ± 6.65 s (~86%), and 35.85 ± 9.21 s (~70%), respectively. All concentrations of the 5-bromocytisine derivatives tested produced an increase in the bottom dwelling time of around ~200 s on average, a value double that of the control.

#### 2.2.2. Average Velocity

The average velocity (Figure 3) of the control group was 5.72 ± 0.45 cm/s. Our results show that with the 3-chlorocytisine derivative at 1 mg/L, the velocity was decreased to 3.082 ± 0.71 cm/s (~46%) and at a concentration of 10 mg/L it decreased to 2.646 ± 0.29 cm/s (~53%); both concentrations were significantly different to the control. The average velocities at 25 mg/L and 50 mg/L were not statistical different to that of the control, giving an average velocity of 5.315 ± 0.31 cm/s and 4.725 ± 0.38 cm/s, respectively. For the 5-chlorocytisine derivative, velocity was decreased at 1mg/L, 10 mg/L and 25 mg/L with average values of 3.512 ± 0.33 cm/s (~38%), 1.599 ± 0.22 cm/s (~72%) and 2.24 ± 0.26 cm/s (~60%), respectively. At a concentration of 50 mg/L, no effect was observed on the average velocity, 6.05 ± 0.41 cm/s. For the 3-bromocytisine derivative, a decrease in velocity was observed at 1 mg/L, 10 mg/L and 25 mg/L with average values of 2.611 ± 0.40 cm/s (~54%), 2.107 ± 0.45 cm/s (63%), and 2.016 ± 0.39 cm/s (~64%), respectively. All of them were significantly different from that of the control. At a concentration of 50 mg/L, the average velocity was 4.746 ± 0.43 cm/s, which is not statistically different to that of the control. All concentrations used for the 5-bromocytisine were not different to those used for the control. 

#### 2.2.3. Freezing

For the control group, the freezing time (Figure 4) was 25.09 ± 4.956 s. For 3-chlorocytisine, the freezing time was 39.83 ± 9.02 s at 1 mg/L, 61.53 ± 7.95 s at 10 mg/L, 20.23 ± 6.98 s at 25 mg/L and 24.13 ± 4.884 s at 50 mg/L; all of them were not significantly different to that of the control except for the 10 mg/L dose where a little increase in the freezing time was observed. A high freezing time was observed at 10 mg/L of the 5-chlorocytisine derivative (163.91 ± 27.36 s); that for the other doses tested, 1mg/L (39.67 ± 7.39 s) and 50 mg/L (29.54 ± 19.85 s), were not different to that of the control and at 25 mg/L a small increase was observed (58.08 ± 16.35 s). For all concentrations of brominated derivatives of cytisine in both positions of bromo substitution, the freezing times were not significant different compared to that of the control at all concentrations tested. Only the 3-bromocytisine at 1mg/L produced a significant increase in the freezing time (83.15 ± 7.25 s). 

#### 2.2.4. Latency to the Top

The time spent by the fish to reach the upper zone was measured (Figure 5); for the control, the latency was 24.63 ± 8.26 s. The 3-chlorocytisine derivative decreased the latency to the top compared to that of the control at a concentration of 1 mg/L and 10 mg/L (9.55 ± 5.96 s and 5.45 ± 3.59 s, respectively), but for 25 mg/L and 50 mg/L the latency was 27.39 ± 9.35 s. and 23.44 ± 8.76 s. Both latencies were not significantly different to that of the control. Briefly, 5-chlorocytisine increased the latency to the top at the 1 mg/L, 25 mg/L and 50 mg/L concentrations (38.88 ± 6.54 s., 35.36 ± 9.65 s., and 51.46 ± 6.98 s., respectively) but only the last one was significantly different to that of the control. At 10 mg/L the 5-chlorocytisine derivative significant decreasing of latency (11.28 ± 5.32 s.). The latency for the bromine derivatives at 3 position shows an important decrease of latency at 10 mg/L, 25 mg/L, and 50 mg/L (4.41 ± 2.32 s., 6.08 ±3.94 mg/L, and 4.33 ± 2.91s., respectively) and were significant different to the control, the lowest concentration (1 mg/L) was not significant different to the control (35.50 ± 12.32 s.). In contrast, 5-bromocytisine produce an increase of the latency at all doses tested. Where, at 1 mg/L (43.18 ± 11.37 s.), 10 mg/L (40.75 ± 9.61 s.), and 25 mg/L (42.95 ± 12.35 s.) were less significant different to the control compare with 50 mg/L (74.64 ± 17.65 s.) that display the highest latency.

## 3. Discussion

In this study we measured the effects of 3-halocytisine and 5-halocytisine derivatives on a zebrafish model of anxiety using the Novel Tank Diving Test paradigm. It has been shown that there is a relationship between the dwelling behavior and the time that the fish spends at the top or bottom of the tank (Figure 1), an anxiolytic effect is observe when they spend less time at the bottom of the tank (↓ time = ↓ anxiety) and an anxiogenic or stressful effect when spending more time at the bottom of the tank (↑ time = ↑ anxiety or stress), Mobility was another factor analyzed on swimming behavior, the absence of movement, also called “freezing”, and it is associated with anxiety or stressful moments, latency to the top, and average velocity also were measured [28,29,31,32].

Our results show that substitution at position 3 by chlorine or bromine produce a decreasing of the bottom dwelling, displaying an anxiolytic effect. The 3-chloro derivative was apparently more potent than the 3-bromo derivative, but at higher concentrations tested (25 mg/L and 50 mg/L) the 3-chlorocytisine shows an opposite effect increasing the time spent at the bottom indicating an anxiogenic profile, our results correlate with other authors that describe an anxiogenic effect of nicotine at higher concentration in rodents [43,44]. The 3-bromocytisine, despite of looks less potent than the 3-chloro derivative, it displays a better anxiolytic range of concentration (10, 25 and 50 mg/L), and shows no anxiogenic profile compare with the 3-chloro derivative at higher concentration. At low concentration (1 mg/L) it does not affect the bottom dwelling compared to the control. Substitution at position 5 was markedly different on the bottom dwelling compare with the position 3. Both derivatives (chlorinated and brominated) show an increase of the time spent at the bottom, thus our results indicate that substitution at position 5 produce an anxiogenic profile at all concentration tested, however, unexpected results with 5-chlorocytisine at 10 mg/L was found, where the bottom dwelling time was significant decreased, producing apparently an anxiolytic effect under the assumption that a decreasing of the bottom dwelling produce an anxiolytic effect. However, this result needs a deeper analysis. Our result shows that substitution of cytisine at position 5 increase the bottom dwelling time (an anxiogenic effect) and the 3 position displays a better anxiolytic profile, where the chlorine derivative appears to be more potent, but the bromine derivative shows a better range of therapeutic effect, without anxiogenic effect at all concentrations tested.

For the 3- and 5-chlorocytisine derivatives average velocity were decreased at 1 mg/L, and 10 mg/L, and additionally the 5-chlorocytisine at 25 mg/L velocity was significant decreased too. For the brominated derivatives, the 3-bromocytisine at 1 mg/L, 10 mg/L and 25 mg/L decreases velocity compared to the control. From literature, there are evidence that some nicotinic drugs, presented anxiolytic effect and at the same time decrease locomotor activity in rodents, such as varenicline or cytisine, both acting as partial agonists of the nAChRs [45,46,47]. In view of these considerations, a little reduction of the velocity will not affect the anxiolytic profile of our drugs.

The average velocity for the 5-bromo derivative was not significant different to the control at all concentration tested, an opposite effect observes compared to the 3-bromo derivative. This observation indicates that bromination of cytisine produce a different stimulus on mobility depending on position substitution, and the size of the bromine atom, since a chlorine (a smaller atom) substitution on 3- and 5- positions affects the average velocity at the same way.

The freezing time that is associated with anxiety and stressful [28] were measured for the halocytisine derivatives, in general, freezing is not affect by the compounds at the different concentration tested compared to the control, only the 3-chlorocytisine at 10 mg/L, 5-cholrocytisine at 25 mg/L and 3-bromocytisine at 1 mg/L produces a little increase of the freezing time. However, the 5-chlorocytisine at 10 mg/L showed a substantial increase of the freezing time, addressing the observation that at this concentration the 5-chloro derivative displayed a decreasing of the bottom time. Our results indicated that a decrease of the bottom dwelling time does not ensure an anxiolytic effect if the freezing time is elevated.

The first exploratory behavior of rats and fish is associated with the time that the animal spend to change the place when is put on a novel environment (the latency), in rats the latency to cross the center place in the open field paradigm is a measure of an anxiolytic effect, similar to rats an anxiolytic index for fish is the latency to move it to the top in the fish tank. For the 3-chlorocytisine a decrease of latency was observe at 1 mg/L and 10 mg/L and for the 3-bromocytisine this decreasing were observe at 1 mg/L, 10 mg/L and 25 mg/L, this data correlate with the observation that this compounds decrease the bottom dwelling time, and not affects the freezing time of the fish, even though the average speed shows a little decrease, the last observation it can explain proposing that the fish feels relaxed. The 5-holacytisine produce an increasing of the latency for all compound, with the exception of the 5-cholorocytisine at 10 mg/L, but in general 5-position is not a chemical pattern for an anxiolytic effect of the halocytisine derivatives, whilst the 3-halo derivatives display good parameters of bottom dwelling time, freezing and latency addressing with an anxiolytic profile on zebrafish, these results are in according to Allan Kalluef & Jonathan Cachat’s “Zebrafish Neurobehavioral Protocols” [29,48].

## 4. Materials and Methods

### 4.1. Animals

Adult zebrafish, both male and female wild strain, were used for the behavioral experiments. They were obtained from the Facultad de Ciencias Biológicas of the Universidad de Concepción, and were kept (5 for each fishbowl in order to avoid stress associated with individual confinement) at 28 °C on a 14:10 light/dark cycle. All experiments were conducted during the light hours. The zebrafish were fed daily with flake tropical fish food “TetraMin Flakes” of Tetra^®^. All fish used were drug and experimentally naïve and were used only once. After the trials, the zebrafish were euthanized using benzocaine and ice-cold water.

### 4.2. Drugs

#### 4.2.1. Isolation of Cytisine from *S. secundiflora*

20.87 g of cytisine was isolated from 4 kg of *S. secundiflora* seeds, the structure was confirmed by one-dimensional 1H-NMR analysis. Briefly, powder seeds were defated using hexane and then extracted with MeOH, the total methanolic extract contain principally alkaloids among other natural products that were not considered for to be isolated. The cytisine from the methanolic extract was purify using a CombiFlash chromatography unit (teledyne ISCO) and chloroform/methanol (90/10) as a liquid phase. Additionally, the crystalline cytisine was obtained from acetone.

#### 4.2.2. Halogenation

##### Chlorination

For the hemisynthesis of 3- and 5-chlorocytisine derivatives, 2 g (10.5 mmoles) of cytisine in acetic acid (10 mL, 60%) was mixed dropwise with a solution of 1.4 g of N-chlorosuccinimide in acetic acid (NClS, AK Scientific, Union City, CA, USA) (10.5 mmoles) at room temperature. After a reflux with stirring for 1 h, the solvent was removed in vacuo at 70 °C, in accordance with and then modified from Imming et al. [40]. The reaction mixture was separated using silica gel and a combiflash chromatography unit and chloroform/methanol (90/10) as a liquid phase. Amounts of 588.7 mg (2.62 mmol, 24.95%) of 3-chlorocytisine and 561.7 mg (2.50 mmol, 23.8%) of 5-chlorocytisine were obtained; using hydrochloric acid (HCl), the purified compound 3- and 5-cholorocytisine derivatives were converted into their hydrochloride salts. Compounds were identified via one-dimensional 1H-NMR using Spectrometer NMR Bruker AvanceTM III 400 in a D_2_O solvent.

##### Bromination

Using conditions similar to those mentioned above [40], 3 g (15.76 mmoles) of cytisine and 2.81 g (15.75 mmoles) of N-Bromosuccinimide (NBS, AK Scientific, Union City, CA, USA) were mixed, 1050.2 mg (3.90 mmol, 24.7%) of 3-bromocytisine and 14,790.0 mg (5.50, mmol, 34.9%) of 5-bromocytisine were obtained, and tartaric acid (C_4_H_6_O_6_) was used to convert the bromocytisine derivatives into the corresponding tartaric salt, which was identified via one-dimensional 1H-NMR using Spectrometer NMR Bruker AvanceTM III 400 in a D_2_O solvent.

### 4.3. Novel Tank Diving Test

The test tank used was an acrylic trapezoid that was 14.5 cm in height, 22 cm for the bottom and 27 cm for the top, with a diagonal of 16 cm, 5 cm wide, and filled with 1.6 L of water. During the experiment, one fish (n = 10 per dose at 1, 5, 10, 25 and 50 mg/L plus controls) was placed in a holding tank for 5 min for acclimation, and then it was immersed in another tank with the dissolved drug (3-halocytisine and 5-halocytisine), or regular water in the case of the control group, for 3 min. Then, it was transferred to a second holding tank for 5 min and, finally, to the test tank where swimming behavior was recorded for 5 min (Figure 2). The whole procedure was carried out in accordance to previous reports [11,33]. For recording purposes, a USB webcam was used, and the tank was backlit with a white acrylic screen to provide enough contrast for detection. The swimming trajectories were analyzed using Noldus Ethovision XT software (Wageningen, The Netherlands). The novel tank diving test was divided into two sections using Ethovision software and the time spent at the bottom section, average velocity, freezing and latency were measured.

### 4.4. Statistical Analysis

The data were analyzed using one-way ANOVA followed by a Dunnett’s multiple comparisons test. The statistical analysis and data visualization were carried out with GraphPad Prism 9.4.1 (Boston, MA, USA) for Windows. The data shown are presented as the mean ± SEM with *p* ≤ 0.05, *p* ≤ 0.01, and *p* ≤ 0.001 regarded as statically significant.

## 5. Conclusions

In this work, we obtained a complete behavioral profile using the NTT at different concentrations of 3-halocytisine and 5-halocytisine derivatives. The time spent at the bottom zone of the fish tank, average velocity, freezing and latency were compared to those of the control at all concentrations tested. The substitution at position 3 showed a better anxiolytic profile displaying a low bottom dwelling time and a little time of latency, and it did not affect the freezing time. Briefly, 3-bromocytisine displayed a better range of therapeutic effects, and 3-chlorocytisine apparently more potently displayed different dose-dependent behaviors at low concentrations showing an anxiolytic effect, but at higher doses displayed the opposite effect. The substitution at position 5 with chlorine or bromine increased the bottom dwelling time and the latency to the top, two parameters that indicate the anxiogenic effects of these compounds. Comparing the 3-bromo and the 5-bromo derivatives, we can observe that only the 3-bromocytisine derivative produced a decrease in the average velocity. This finding shows that locomotor activity (velocity) depends on the position of substitution for this type of derivative. Unexpected results were found with 3-cholorocytisine at 10 mg/L, these being a significant decrease in the bottom dwelling time and a high freezing time.

In conclusion, our results show that substitutions with chlorine and bromine in position 3 display an anxiolytic profile in the NTT. However, the 3-brominated derivative appeared less potent but without the anxiogenic effects seen with higher concentrations of 3-chloro derivatives. In contrast, position 5 did not allow an anxiolytic effect, thus indicating an anxiogenic profile upon the novel tank diving test.

Regarding the binding affinities and the functional potencies and efficacies of the 3-halo derivatives compared to those of the 5-halo derivatives, our results correlate with the in vitro data published, and show that 3-chloro and 3-bromo derivatives are able to display an anxiolytic effect in the NTT paradigm but that the 5-halo derivatives do not show the same effect. In addition, we highlight the novel tank diving test (NTT) as a powerful tool for screening drugs to investigate their potential anxiolytic activity.

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
