# Peer review of "Behavioral Study of 3- and 5-Halocytisine Derivatives in Zebrafish Using the Novel Tank Diving Test (NTT)"

_ijms, 2023, doi:10.3390/ijms241310635_

Round 1
Reviewer 1 Report
Overall, the work was very well designed experimentally.
The introduction includes relevant and up-to-date information on the topic addressed. The question of the increasing prevalence of anxiety-like behavior worldwide is a current problem and studies that seek treatment alternatives are necessary and relevant.
The use of the Zebrafish model is very interesting and has been shown to be adequate in translational studies.
In light of the foregoing, the present work intended to study the anxiolytic effect of derivative compounds of Cytisine, an alkaloid present in the plant of the genus Sophora, with potential agonist of nicotinic receptors. Such receptors, such as the Nicotinic Acetylcholine Receptors (nAChRs), have already been associated with an anxiety-related mechanism. Here, the group tested different doses of 4 derivative compounds (3- and 5-halocytisine derivatives) and their anxiolytic potential. To do so, they chose the adult Zebrafish model and analyzed the paradigm of the Novel Tank Diving Test (NTT). The NTT is already well established in the literature as being an adequate test for the evaluation of anxiety-like behavior in this animal model. Through this methodology, the authors managed to arrive at a promising result, listing the 3-chloro derivative, at a concentration of 1mg/L with the lowest time at the bottom of the tank and therefore with a promising anxiolytic effect.
The results were displayed clearly and very well organized.
The discussion deserves to be praised, especially the first paragraph, which was positively illustrated by scheme 1.
Perhaps the only drawback of the work was the lack of molecular methodology, for example, quantification of Nicotinic Acetylcholine Receptors (nAChRs) transcripts and their subtypes. What will be the effect of the derivatives on the transcriptional regulation of these receptors? And what about proteins? The same question could be asked. Anyway, and according to the title of the manuscript, the authors aimed to analyze the behavior of animals when treated with different compounds derived from Cytisine. In this way, the objective was achieved.
As for self-citation, we found 5 references of works published by the authors, in a total of 49 references. For example, reference 33, “Effects of Selective Α4β2 Nicotinic Acetylcholine Receptor (NAChR) Ligands on the Behavior of Adult Zebrafish (Danio Rerio) in the Novel Tank Diving Task”. Reference, 35, same thing. All self-citations are significant for the discussion of this work and, therefore, necessary, and relevant.
Reference number 50 does not exist and must be removed from the text.
Finally, the work is relevant, very well prepared and developed and deserves to be published.
Author Response
Many thanjs for the commentaries about our work, and we response to the molecular biology question
Perhaps the only drawback of the work was the lack of molecular methodology, for example, quantification of Nicotinic Acetylcholine Receptors (nAChRs) transcripts and their subtypes. What will be the effect of the derivatives on the transcriptional regulation of these receptors? And what about proteins?
We agree with the referee comments and we have at this time detecting nicotinic subunits genes and we are working on protein expression, but we have not enough results for to publish
Reviewer 2 Report
In present research, adult zebrafish was used to determine the behavioral effects of 3 - and 5-halocytisine derivatives, using the NTT at different doses. The results show that substitution at position 3 by chlorine or bromine decreases the time spent by the fish at the bottom compared to the control. However, the 3-chloro derivative at higher doses increases the bottom-dwelling time. In contrast, substitution at 5 position increases the bottom dwelling at all concentrations showing no anxiolytic effects in this model. Unexpected results were observed with the 5-chlorocytisine derivative, at a concentration of 10 mg/L produced a significant decrease of the bottom-dwelling, and shown high times of freezing. The 3-chloro and 3-bromo derivatives show an anxiolytic effect, being the 3-chlorocytisine more potent than the 3-bromo derivative, with the lowest time at the bottom of the tank at 1mg/L. On the other hand, chlorine and bromine at 5 position produce an opposite effect.
Why the authors have chosen 3- and 5-halocytisine derivatives for this research? It should be more explained at the introduction part.
At the end of introduction part, the main aim was described. However, there is need to point out more the novelty of present research. What was done for the first time? The novelty should be stated more precise.
Overall paragraph “Our results show that substitution at position 3 by chlorine or bromine decreases the time spent by the fish at…” and ending with “On the other hand, chlorine and bromine at 5 positions produce an opposite effect increasing the bottom-dwelling time” is redundant and should be deleted. In the introduction part there is no place to state the obtained results.
Figure 5. should be moved at the position in the introduction after mentioning them in the text for the first time.
2.2. Drugs – it was written plural for NMR analysis and only cysteine was isolated. It should be corrected.
4.2.1. Isolation of cytisine from S. secundiflora - “Briefly, powder seeds were defeated using hexane and then extracted with MeOH until total extraction of alkaloids.” Are the authors sure that MeOH is extracting only alkaloids? It should be revised.
4.2.2.1. Chlorination and 4.2.2.2. Bromination – were these reactions performed according to some reference? The literature should be added for these reactions.
At the end of conclusions there is need to state what are further potential for this research.
-
Author Response
Thanks for the feedback about our work, we have made all the suggested changes, see answers below, and the changes are highlighted in red in the manuscript.
Why the authors have chosen 3- and 5-halocytisine derivatives for this research? It should be more explained at the introduction part.
We use the 3- and 5-halocytisine since synthetic methodology and the reactivity of the pyridone functional group allows substitution only on these positions, the 3,5-halocytisine was not isolated in good yields to perform the behavioural experiments.
At the end of introduction part, the main aim was described. However, there is need to point out more the novelty of present research. What was done for the first time? The novelty should be stated more precise.
We add two new paragraph at the end of the introduction
These compounds have shown a high affinity (sub nM) for the nAChR subtypes, the 3-bromo derivative display selectivity for the heteromeric a4b2 (IC50 = 0.30 nM) and a4b4 (IC50 = 0.28 nM) nAChR over the homomeric a7 subtype (IC50 = 31 nM). Other positions, the 5-halo and the 3,5-dihalo derivatives displays low affinities in the microM range
Based on the data that shows different affinities and potencies of the halo-cytisine derivatives, and points out to the 3-halo derivatives as the most potent compared with the 5 position, the main aim of this work was to determine and compare the anxiolytic effect of 3-, and 5-halo-cytisine derivatives (Figure 1) using the novel tank diving test (NTT) as an anxious paradigm and to confirm that nicotinic partial agonists exert anxiolytic effects in this model.
Overall paragraph “Our results show that substitution at position 3 by chlorine or bromine decreases the time spent by the fish at…” and ending with “On the other hand, chlorine and bromine at 5 positions produce an opposite effect increasing the bottom-dwelling time” is redundant and should be deleted. In the introduction part there is no place to state the obtained results.
This paragraph was deleted.
Figure 5. should be moved at the position in the introduction after mentioning them in the text for the first time.
Was Changed
2.2. Drugs – it was written plural for NMR analysis and only cysteine was isolated. It should be corrected.
Was changed
4.2.1. Isolation of cytisine from S. secundiflora - “Briefly, powder seeds were defeated using hexane and then extracted with MeOH until total extraction of alkaloids.” Are the authors sure that MeOH is extracting only alkaloids? It should be revised.
Briefly, powder seeds were defated using hexane and then extracted with MeOH, the total methanolic extract contain principally alkaloids among other natural products that were not considered for to be isolated
4.2.2.1. Chlorination and 4.2.2.2. Bromination – were these reactions performed according to some reference? The literature should be added for these reactions.
Reference was added to the synthetic methodology
At the end of conclusions there is need to state what are further potential for this research.
We improve the final part of the conclusion with a new paragraph.
Regarding the binding affinities and the functional potencies and efficacies of the 3-halo derivatives compared with the 5-halo derivatives, our results correlate with the in vitro data published, and shows that 3-chloro and 3-bromo derivatives are able to display an anxiolytic effect on the NTT paradigm but the 5-halo derivatives do not show the same effect, in addition, highlight to the novel tank diving test (NTT) as a powerful tool for screening drugs searching for its potential anxiolytic activity.